

# A novel fully convolutional network for visual saliency prediction

Bashir Muftah Ghariba[1,2], Mohamed S. Shehata[3] and Peter McGuire[4]

[1] Faculty of Engineering & Applied Science, Memorial University of Newfoundland, St. John's, NL, Canada
[2] Department of Electrical and Computer Engineering, Faculty of Engineering, Elmergib University, Khoms, Libya
[3] Department of Computer Science, Mathematics, Physics and Statistics, University of British Columbia, Kelowna, BC, Canada
[4] C-CORE, St John's, NL, Canada

## ABSTRACT

A human Visual System (HVS) has the ability to pay visual attention, which is one of the many functions of the HVS. Despite the many advancements being made in visual saliency prediction, there continues to be room for improvement. Deep learning has recently been used to deal with this task. This study proposes a novel deep learning model based on a Fully Convolutional Network (FCN) architecture. The proposed model is trained in an end-to-end style and designed to predict visual saliency. The entire proposed model is fully training style from scratch to extract distinguishing features. The proposed model is evaluated using several benchmark datasets, such as MIT300, MIT1003, TORONTO, and DUT-OMRON. The quantitative and qualitative experiment analyses demonstrate that the proposed model achieves superior performance for predicting visual saliency.

## INTRODUCTION

A Human Visual System (HVS) processes a part of the visual scene instead of the whole scene. This phenomenon is called Human Visual Attention (HVA), also referred to as visual saliency prediction, which is an important research area in the field of computer vision. HVA is also known as human eye fixation prediction, visual saliency prediction, or saliency map detection. Visual saliency prediction is also beneficial for other applications in the computer vision field, including salient object detection (*Liu & Han, 2016*), image retrieval (*Huang et al., 2011*), multiresolution imaging (*Lu & Li, 2013*), and scene classification (*Cheng et al., 2015*; *Lu, Li & Mou, 2014*; *Yao et al., 2016*).

Many models have been developed to predict visual saliency, the most popular being the saliency map. Saliency maps describe the probability that each image pixel will attract human attention. In other words, saliency maps are images that display the unique qualities of each pixel in a given image (*Gao & Vasconcelos, 2005*). To produce a saliency map, the salient points in the image are collected and convolved with a Gaussian filter (*Gao & Vasconcelos, 2005*). The probability that each pixel in the image will attract human attention is represented by a heat map or gray-scale image. Notably, saliency maps smooth

Corresponding author
Bashir Muftah Ghariba,
bmg063@mun.ca

the image, making it more meaningful and easier to analyze. This is useful for condition image captioning architecture because it indicates what is salient and what is not (*Mackenzie & Harris, 2017*).

To evaluate the saliency map, human eye fixation data in free viewing is used because there is a direct link between human eye movement and visual attention (*Mackenzie & Harris, 2017*). Generally, HVA runs on two approaches. The first is a bottom-up approach which utilizes low-level features, including intensity, color, edge orientation, and texture (*Gao, Mahadevan & Vasconcelos, 2008*; *Le Meur et al., 2006*). Such an approach attempts to decide regions that show obvious characteristics of their surroundings. The second is a top-down approach, which is task-driven and requires an explicit understanding of the context of the visual scene. Moreover, it depends on the features of the object of interest (*Gao, Han & Vasconcelos, 2009*; *Kanan et al., 2009*).

The deep Convolutional Neural Network (CNN) is the most widely utilized deep learning method for image processing applications (*Mahdianpari et al., 2018*). Specifically, CNN is capable of extracting discriminant visual features (e.g., 2-D spatial features) by applying a hierarchy of convolutional filters using multiple nonlinear transformations. Studies have also used Convolutional Neural Networks (CNNs) for studying saliency map detection to confirm the importance of end-to-end task learning and automatic feature extraction (*Fang et al., 2016*; *Jetley, Murray & Vig, 2016*; *Kruthiventi et al., 2016*; *Pan et al., 2016*; *Vig, Dorr & Cox, 2014*). The deep CNN model achieves an even higher classification accuracy. For example, deep learning techniques have achieved superior results in multiple tasks, such as driverless car, scene classification, object (e.g., vehicle) detection, image classification, and semantic segmentation. However, deep learning architecture requires sufficient training data for superior performance on several sets of visual tasks, such as local image detection (*Girshick et al., 2014*), global image classification (*Krizhevsky, Sutskever & Hinton, 2012*), and semantic segmentation (*Long, Shelhamer & Darrell, 2015*).

Although several deep learning models have been proposed to solve the problem of saliency prediction, and those models provide good performance. However, those models essentially proposed for object recognition and then fine-tuned for saliency prediction. Consequently, the pixel-based classification of visual attention task remains challenging. This highlights the necessity of designing a novel FCN model specifically for the task of saliency prediction. In addition, our proposed model is designed for training from scratch. Therefore, we added some modules (e.g., three inception modules and residual modules) to improve the model performance.

The inception module is useful since benefits from filters with different sizes in one layer, which contribute to multi-scale inference and enhance contextual information. This highlights the necessity of combining feature maps at different resolution to extract useful information. In addition the residual module recovers more accurate information and simplifies optimization, while avoiding the vanishing gradient problem.

In this study, we utilized an encoder–decoder structure based on the Fully Convolutional Network (FCN) architecture to address the problem of bottom-up visual attention in visual saliency predication. FCN has the same architecture as the CNN network, but unlike CNN it does not contain any fully connected layers. FCNs are also powerful visual models that

generate high-level features from low-level features to produce hierarchies. Moreover, FCN utilizes multi-layer information and addresses pixel-based classification tasks using an end-to-end style (*Long, Shelhamer & Darrell, 2015*). In addition, the proposed model also includes both inception and residual modules to improve multi-scale inference and the recovery of more accurate information, respectively.

This study proposes a new model based on an encoder–decoder structure (i.e., FCN) to improve the performance of visual saliency prediction. The specific contributions of this work are as follows:

(1) A new model of FCN architecture for visual saliency prediction that uses two types of modules is proposed. The first module contains three stages of inception modules, improves the multi-scale inference, and performs contextual information. The second module contains one stage from the residual module and also recovers more accurate information and simplifies optimization, while avoiding the vanishing gradient problem.

(2) Four well-known datasets, including TORONTO, MIT300, MIT1003, and DUT-OMRON, were used to evaluate the proposed model. The experiments demonstrate that the proposed model achieves results comparable or superior to those of other state-of-the-art models.

The remainder of this article is organized as follows. First, the proposed model is described in more detail in 'Related Work' and the materials and methods used to produce and evaluate the proposed model are discussed in 'Material and Methods'. 'Experimental Results' presents the quantitative and qualitative experimental results obtained from the four datasets. Finally, the results are summarized, and possible future uses and applications of the proposed model are explored in 'Discussion'.

## RELATED WORK

Visual saliency prediction has received attention from computer vision researchers for many years. The earliest computational model was introduced by Koch and Ullman (*Krizhevsky, Sutskever & Hinton, 2012*), which inspired the work of *Itti, Koch & Niebur (1998)*. This model combines low level features at multiple scales to generate saliency maps. Subsequently, many models have been proposed to address visual saliency detection (*Fu et al., 2015*; *Gong et al., 2015*; *Guo et al., 2017*; *Li et al., 2014*; *Liu et al., 2016*; *Liu et al., 2014a*; *Liu, Zou & Meur, 2014b*; *Wang & Shen, 2017*; *Wang et al., 2019a*; *Wang et al., 2016*; *Wang et al., 2017b*). Most of this work has been focused on how to detect visual saliency in an image/video using different methods (*Borji & Itti, 2012*; *Wang, Shen & Shao, 2017a*; *Wang et al., 2019b*).

Most conventional attention models are based on a bottom-up strategy. These contain three important steps to detect visual saliency: feature extraction, saliency extraction, and saliency combination. Salient regions in the visual scene are first extracted from their surroundings through hand-crafted low-level features (e.g., intensity, color, edge orientation, and texture), and center–surround contrast is widely used for generating saliency. The saliency may also be produced by the relative difference between the region and its local surroundings (*Itti, Koch & Niebur, 1998*; *Harel, Koch & Perona, 2007*; *Bruce &*

*Tsotsos, 2006*). The last step for saliency detection combines several features to generate the saliency map.

Recently, many visual saliency models have been introduced for object recognition. Deep-learning models achieved better performance compared to non-deep learning models. The Deep Neural Networks (DNN) (*Vig, Dorr & Cox, 2014*), was trained from scratch to predict saliency. Subsequent models were based on pre-trained models, for example, the DeepGaze I model (*Kümmerer, Theis & Bethge, 2014*), which was the first to be trained on a pre-trained model (AlexNet *Krizhevsky, Sutskever & Hinton, 2012*) trained on ImageNet (*Deng et al., 2009*), and outperformed the training stage from scratch. DeepGaze II (*Kümmerer, Wallis & Bethge, 2016*) has also has been proposed based on a pre-trained model (VGG-19 *Simonyan & Zisserman, 2014*), where attention information was extracted from the VGGNet without fine-tuning the attention task. Next, the DeepFix model *Kruthiventi, Ayush & Babu (2017)* was proposed by Kruthiventi et al. based on a pre-trained model VGG-16. Furthermore, in *Mahadevan & Vasconcelos (2009)* object detection and saliency detection were carried out using a deep convolutional neural network (CNN). Finally, the SALICON net model (*Huang et al., 2015*) was proposed to capture multi-scale saliency using combined fine and coarse features from two-stream CNNs that were trained with multi-scale inputs.

Since the superior success of transfer learning models for visual saliency prediction has been established, several new models have been proposed that have improved saliency prediction performance. For instance, the SALICON model fine-tunes a mixture of deep features (*Huang et al., 2015*) using AlexNet (*Krizhevsky, Sutskever & Hinton, 2012*), VGG-16 network (*Simonyan & Zisserman, 2014*), and GoogleNet (*Szegedy et al., 2015*) for visual saliency prediction. PDP (*Jetley, Murray & Vig, 2016*) and DeepFix (*Kruthiventi, Ayush & Babu, 2017*) were used on the VGG-19 network for the same task using MIT300 and the SALICON dataset, and FUCOS (*Bruce, Catton & Janjic, 2016*) fine-tunes features that were trained on the PASCAL dataset. Overall, DeepFix and SALICON models demonstrated significantly improved performance compared to DeepGaze I in the MIT benchmark.

## MATERIAL AND METHODS

### Proposed model

The proposed model follows an FCN structure (i.e., a pixel-based approach) and the generic encoder–decoder form. The important difference between CNN and FCN networks is that the latter has learning filters throughout its structure. Even the decision-making layers at the end of the network are filters. FCNs also do not have any fully connected layers that are usually available at the end of the network.

Figure 1 explains the architecture of the proposed model for visual saliency prediction and the configuration of the proposed model is explained in Table 1. The encoder stage contains three blocks of convolution layers, each of which is followed by batch normalization, rectified linear unit (ReLU), and max pooling. The encoder stage is the same as that of a conventional CNN and generates feature maps by down-sample pooling. The decoder stage also transposes convolutional layers but does so in the opposite direction.

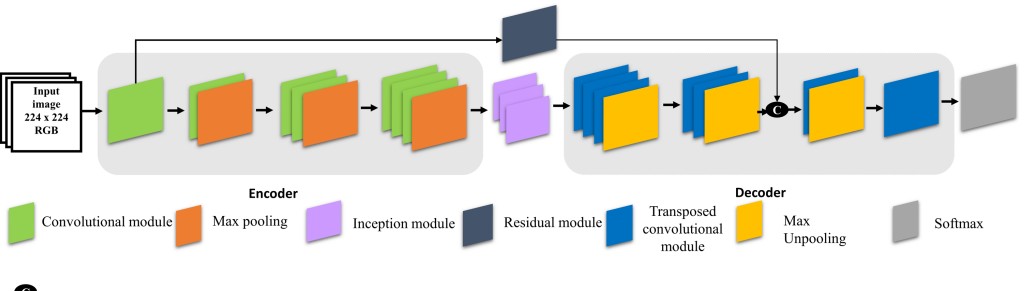

**Figure 1  Architecture of the proposed model.**

**Table 1  Configuration of the proposed model.**

|  | Layer type | Filter size |
|---|---|---|
| **Encoder** | Convolution | $3 \times 3$, 64 |
|  | Residual Module | (*), 64 |
|  | Convolution | $3 \times 3$, 128 |
|  | Max pooling | $2 \times 2$ |
|  | Convolution | $3 \times 3$, 256 |
|  | Max pooling | $2 \times 2$ |
| **Decoder** | Inception Module | (*), 256 |
|  | Transposed convolution | $3 \times 3$, 256 |
|  | Convolution | $3 \times 3$, 256 |
|  | Convolution | $3 \times 3$, 128 |
|  | Transposed convolution | $3 \times 3$, 64 |
|  | Convolution | $3 \times 3$, 2 |
|  | Pixel Classification Layer | — |

Therefore, the decoder stage produces label maps (up-sampling) with the same input image size. The transposed convolution layers contain un-pooling and convolution operators. Unlike the max-pooling operation, the un-pooling operation increases the size of feature maps through the decoding stage. In addition, the image input size of the proposed model is 224 × 224 pixels. Figure 1 illustrates the proposed model architecture to predict visual saliency.

Three inception modules are also used in the proposed model. Inception modules are useful because they benefit from different sized filters in one layer, which contributes to the multi-scale inference and enhances contextual information (*Long, Shelhamer & Darrell, 2015*). In addition, a residual module is also added to the proposed model because it effectively avoids the vanishing gradient problem by introducing an identity shortcut connection (*Lin et al., 2014*). Moreover, activations from a previous layer are reused by the residual module for the adjacent layer to learn its weights. Figure 2 shows the architecture of the inception and residual modules, respectively. Figure 2A explains the layers of the inception module which contains three branches. The first two contain a sequence of two convolution filters, where the patch sizes of the layers are 1 × 1, the second layer is 3 × 3, and

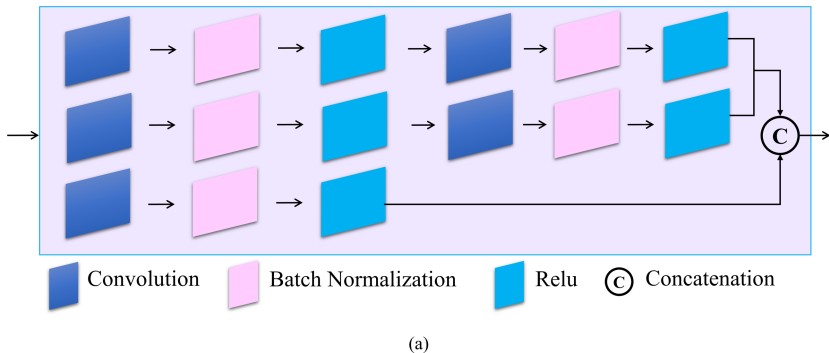

Convolution    Batch Normalization    Relu    C Concatenation

(a)

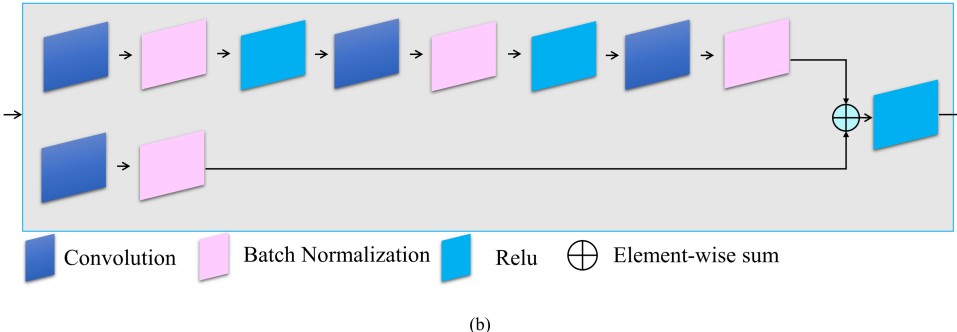

Convolution    Batch Normalization    Relu    ⊕ Element-wise sum

(b)

**Figure 2**   **Architecture of (A) Inception and (B) residual modules.**

**Table 2**   **Configuration of inception and residual modules.**

| Module | Convolutional configuration | | | Operation | Output |
|---|---|---|---|---|---|
| Inception | 1 × 1,256<br>1 × 1,256<br>1 × 1,256 | 3 × 3,256<br>5 × 5,256 | | Concatenation | 256 |
| Residual | 1 × 1,32<br>1 × 1,64 | 3 × 3,64 | 1 × 1,64 | Element-wise sum | 64 |

the last layer is 5 × 5, respectively. The third branch contains only one convolutional filter which has a patch size of 1 × 1. Each convolutional layer is followed by batch normalization and ReLU. Figure 2B explains the structure of the residual module, which contains two branches. The first branch has a stack of three convolutional filters, sized 1 × 1, 3 × 3, and 1 × 1, respectively. The second branch has a single 1 × 1 convolutional filter. The two branches are combined by element-wise summation. Table 2 explains the number of each filter in the two modules (i.e., inception and residual). Notably, the convolutional module contains a Convolutional 2D, Batch Normalization, as well as a ReLU layer. The transposed convolutional module also contains the same layers as the convolutional module.

## Semantic segemnation

The segmentation task plays an important role in image understanding and is essential for image analysis tasks (*Karami, Shehata & Smith, 2018*). In semantic segmentation, each region or pixel is labeled as a class, such as flower, person, road, sky, ocean, or car. Many applications use semantic segmentation techniques, such as autonomous driving, Bio Medical Image Diagnosis, robotic navigation, localization, and scene understanding. Furthermore, Deep Neural Networks (DNNs) are commonly used as effective techniques for semantic segmentation (*Long, Shelhamer & Darrell, 2015*). Semantic segmentation works with semantics and location; global information determines the "what" while local information determines the "where" of an image. Deep feature hierarchies encode semantics and location in a nonlinear local-to-global pyramid (*Long, Shelhamer & Darrell, 2015*). Our proposed model (i.e., FCN) uses semantic segmentation techniques to assign each pixel in the given image into appropriate classes (i.e., foreground or background) in order to predict visual saliency (i.e., saliency map generation).

## Datasets

The proposed model was trained using a standard available dataset (i.e., SALICON) and subsequently tested on four other well-established datasets, including TORONTO, MIT 300, MIT1003, and DUT-OMRON. All these datasets have different characteristics and so each is described below.

### SALICON

The largest dataset for visual attention applications on the popular Microsoft Common Objects in Context (MS COCO) image database is SALICON (*Lin et al., 2014*). This dataset contains 10,000 training, 5,000 validation, and 5,000 testing images with a fixed resolution of 480x640. While this dataset contains the ground truth data for the training and validation datasets, the test dataset ground truth data were unavailable (*Jiang et al., 2015*).

### TORONTO

One of the most widely used datasets for visual attention is the TORONTO dataset. It has 120 color images with a resolution of 511x681 pixels. This dataset contains images that were captured in indoor and outdoor environments and has been free-viewed by 20 human subjects (*Bruce & Tsotsos, 2006*).

### MIT300

The MIT300 dataset has 300 natural images and the eye-tracking data of 39 users who free viewed these images were used to generate saliency maps. This dataset is challenging since its images are highly variable and natural (*Judd, Durand & Torralba, 2012*). A MIT saliency benchmark website for model evaluation (http://saliency.mit.edu/results_mit300.html) is available to evaluate any saliency model using this dataset.

### MIT1003

MIT1003 includes 1,003 images from the Flicker and LabelMe collections. Saliency maps of these images have also been generated from the eye-tracking data of 15 users. This dataset

contains 779 landscape and 228 portrait images that vary in size from 405 × 405 to 1,024 × 1,024 pixels, making it the largest available eye fixation dataset (*Judd et al., 2009*).

### DUT-OMRON

DUT-OMRON has 5,168 high quality images that were manually selected from over 140,000 images. The largest height or width of this dataset is 400 pixels and each image is represented by five subjects. There is more than one salient object in this type of dataset and the image has a more a complex background (*Riche et al., 2013*).

## Evaluation metrics

Several methods may be used to evaluate the correspondence between human eye fixation and model prediction (*Ghariba, Shehata & McGuire, 2019*). Generally, saliency evaluation metrics are divided into distribution- and location-based metrics. Previous studies on saliency metrics found it is difficult to perform a reasonable comparison for assessing saliency models using a single metric (*Riche et al., 2013*). Here, we accomplished our experiment by extensively considering several different metrics, including the Similarity Metric (SIM), Normalized Scanpath Saliency (NSS), and AUC. The last metric is the area under the receiver operating characteristic (ROC) curve (e.g., AUC-Borji, and AUC-Judd). For clarification, we indicate the map of fixation locations as Q, the predicted saliency map as S, and the continuous saliency map (distribution) as G.

### Similarity Metric (SIM)

The SIM metric produces a histogram that is a measurement of the similarity between two distributions. This metric considers the normalized probability distributions of both the saliency and human eye fixation maps. SIM is also computed as the sum of the minimum values at each pixel, after normalizing the input maps. Equation (1) explains how to calculate the SIM metric.

$$\text{SIM} = \sum_{i=1} \min(\acute{S}(i), \acute{G}(i)), \tag{1}$$

where $\sum_i \acute{S}(i) = 1$, and $\sum_i \acute{G}(i) = 1$, and $\acute{\mathbf{S}}$ and $\acute{\mathbf{G}}$ are the normalized saliency and the fixation maps, respectively. Importantly, a similarity of one indicates that the distributions are the same whereas a zero indicates that they do not overlap.

### Normalized Scanpath Saliency (NSS)

NSS was is a simple correspondence measure between saliency maps and ground truth data, computed as the average normalized saliency at fixated locations. NSS is, however, susceptible to false positives and relative differences in saliency across the image (*Bylinskii et al., 2018*). To calculate NSS given a saliency map S and a binary map of fixation location F,

$$\text{NSS} = \frac{1}{N} \sum_{i=1}^{N} \bar{S}(i) \times F(i), \tag{2}$$

where $N = \sum_i F(i)$ and $\bar{S} = \frac{S - \mu(s)}{\sigma(S)}$, and N is the total number of human eye positions and $\sigma(S)$ is the standard deviation.

**Table 3  Comparison of the quantitative scores of several models on the TORONTO (*Bruce & Tsotsos, 2006*) dataset.**

| Model | NSS | SIM | AUC-Judd | AUC-Borji |
|---|---|---|---|---|
| ITTI | 1.30 | 0.45 | 0.80 | 0.80 |
| AIM | 0.84 | 0.36 | 0.76 | 0.75 |
| Judd Model | 1.15 | 0.40 | 0.78 | 0.77 |
| GBVS | 1.52 | 0.49 | 0.83 | 0.83 |
| Mr-CNN | 1.41 | 0.47 | 0.80 | 0.79 |
| CAS | 1.27 | 0.44 | 0.78 | 0.78 |
| Proposed Model | 1.52 | 0.46 | 0.80 | 0.76 |

### *AUC-Borji*

The AUC-Borji metric, based on Ali Borji's code (*Borji et al., 2013*), uses a uniform random sample of image pixels as negatives and defines false positives as any fixation (saliency) map values above the threshold of these pixels. The saliency map is a binary classifier that separates positive from negative samples at varying thresholds, the values of which are sampled at a fixed step size. The proportion of the saliency map values above the threshold at the fixation locations is the true positive (TP) rate. Conversely, the proportion of the saliency map values that occur above the threshold sampled from random pixels (as many samples as fixations, sampled uniformly from all image pixels) is the false positive rate (FP).

### *AUC-Judd*

The AUC-Judd metric (*Judd et al., 2009*) is also popular for the evaluation of saliency models. As with AUC-Borji, positive and negative samples are separated at various thresholds by treating the saliency map as a binary classifier. Unlike AUC-Borji, however, the thresholds are sampled from the saliency map's values. The proportion of the saliency map's values above a specific threshold at specific fixation locations is known as the true positive (tp) rate. Alternatively, the proportion of the saliency map's values that occur above the threshold of non-fixated pixels is the false positive (fp) rate.

## EXPERIMENTAL RESULTS

This Section explains all the steps for implementing our work (see Table 3 for more details about experimental steps). Specifically, training, adjusting the parameters, validating, and testing the proposed model on the aforementioned datasets (e.g., TORONTO, MIT300, MIT1003, and DUT-OMRON) are described in details.

### Model training

The most important step for the proposed model is model training. In this work, the proposed model was trained from scratch (i.e., full-training). Training of models from scratch is challenging due to computational and data availability, leading to problems of overfitting. However, there are several techniques, such as normalization, data augmentation, and dropout layers that are useful for mitigating the problems generated from overfitting.

The full-training style has two different categories. In the first category, the CNN architecture is fully designed and trained from scratch. In this case, the number of CNN, pooling layers, the kind of activation function, neurons, learning rate, and the number of iterations should be determined. In the second category, the network architecture and the number of parameters remain unchanged, but the advantages of pre-existing architecture and full-training is applied to given images.

In this study, the first category was employed. Specifically, the proposed model was trained using the well-known dataset, SALICON (see 'SALICON' for more details) and was also validated using a specific validation dataset (i.e., 5000 images). This dataset is the largest available for visual attention (i.e., 10,000 images for training, 5,000 for validation) and was created for saliency applications. At the beginning of the training task, all filter weights were randomly initialized because a pre-trained network was not used in this study. A mini-batch of 16 images was used in each iteration and the learning rate was set as 0.001. The proposed model parameters were learned using the back-propagating loss function by stochastic gradient descent with a momentum (SGDM) optimizer.

Since the number of images available for the training task was limited (i.e., 10,000 images), we suggested using the date augmentation technique to increase the number of training images by creating modified versions of images in the dataset. This technique was carried out to mitigate overfitting by rotating at 30° intervals. This technique also improves performance and the proposed model's ability for generalization. Figure 3 illustrates the proposed model's training progress from the mentioned training images (SALICON).

## Model testing

In this step, we evaluated the proposed model using very well-known datasets including TORONTO, MIT300, MIT1003, and DUT-OMRON. Based on the experimental results, one can see the proposed model has the ability to predict visual saliency in a given image. The output of the test image is described as the saliency map, which can be obtained from the last layer of the proposed model. All the training and testing tasks were performed on an Intel CPU i7-3370K machine with 3.5 GHz and 16 GB RAM memory. An NVIDIA GeForce GTX 1080 Ti GPU with 11 GB of memory under CUDA version 8.0 was also utilized in this work.

## DISCUSSION

### Quantitative comparison of the proposed model with other advanced models

To evaluate the efficiency of the proposed model for predicting visual saliency, we compared it to 10 state-of-the-art models, ITT, AIM, Judd Model, GBVS, Mr-CNN, CAS, SalGAN, DeepGaze I, DeepGaze II, and ML-NET. The models were applied to four datasets (i.e., TORONTO, MIT300, MIT1003, and DUT-OMRON), and the quantitative results are presented in Tables 3, 4, 5 and 6, respectively. All these models differ in terms of computational speed (i.e., run time). Table 7 explains the runtime properties of the proposed model as well as the other 10 visual saliency models. From this table, one can see

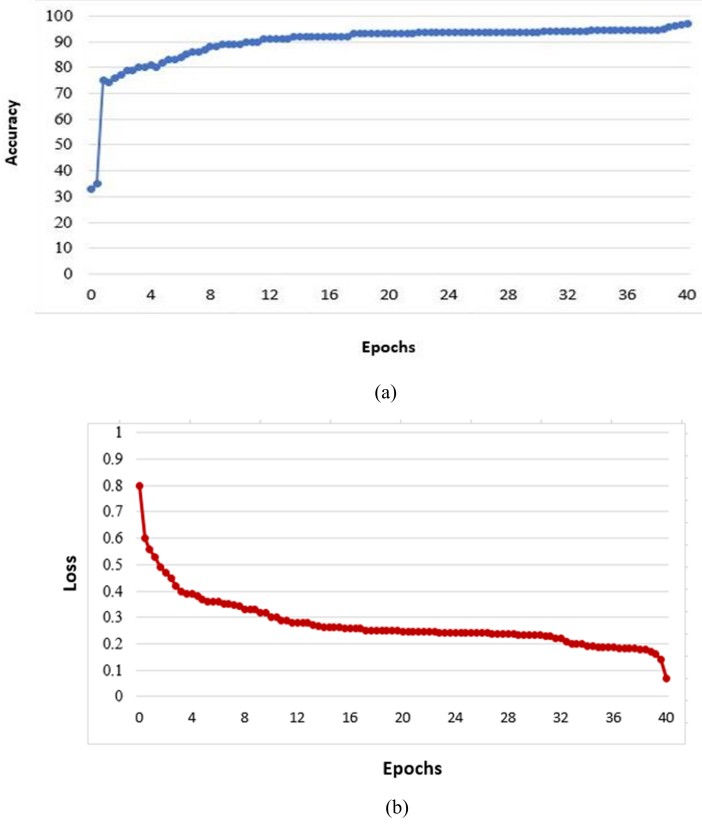

(a)

(b)

**Figure 3** Value of validation accuracy (A) and loss as a function of epochs (B).

the run time of the proposed model is about 12 s on our machine which has no GPU and is an Intel CPU i7-3370K.

Notably, the main difference between the proposed model and other state-of-art models is that the proposed model was specifically designed for saliency prediction, whereas the other pre-trained models were essentially designed for object recognition and then fine-tuned for the visual saliency prediction task. In addition, the proposed model was trained from scratch, which requires a large number of training images to provide a reasonable performance; however, the largest dataset available for this application contains only 10,000 images (e.g., SALICON), which is considered relatively small to train a model from scratch.

Table 3 shows that, with the TORONTO dataset, the proposed model outperforms other models (deep and classical models) in terms of NSS; however, in terms of SIM, AUC-Judd, and AUC-Borji, the GBVS model provides the best results (note that the bolded values are the best results). From Table 4, one can see that with the MIT300 dataset, the model that provides the best performance is DeepGaz II in terms of the AUC-Judd and AUC-Borji metrics. However, the SalGAN model produces the best results for the SIM metric, while the ML-NET model provides the best value for the NSS metric. In Table 5 (for MIT1003 dataset), one can see that the proposed model surpasses the other models in terms of the

**Table 4** Comparison of the quantitative scores of several models on the MIT300 (*Judd, Durand & Torralba, 2012*) dataset.

| Model | NSS | SIM | AUC-Judd | AUC-Borji |
|---|---|---|---|---|
| ITTI (*Itti, Koch & Niebur, 1998*) | 0.97 | 0.44 | 0.75 | 0.74 |
| AIM (*Bruce & Tsotsos 2006*) | 0.79 | 0.40 | 0.77 | 0.75 |
| Judd Model (*Judd et al., 2009*) | 1.18 | 0.42 | 0.81 | 0.80 |
| GBVS (*Harel, Koch & Perona, 2007*) | 1.24 | 0.48 | 0.81 | 0.80 |
| Mr-CNN (*Liu et al., 2016*) | 1.13 | 0.45 | 0.77 | 0.76 |
| CAS (*Goferman, Zelnik-Manor & Tal, 2011*) | 0.95 | 0.43 | 0.74 | 0.73 |
| SalGAN (*Pan et al., 2016*) | 2.04 | 0.63 | 0.86 | 0.81 |
| DeepGaze I (*Kümmerer, Theis & Bethge, 2014*) | 1.22 | 0.39 | 0.84 | 0.83 |
| DeepGaze II (*Kümmerer et al., 2017*) | 1.29 | 0.46 | 0.87 | 0.86 |
| ML-NET (*Cornia et al., 2016*) | 2.05 | 0.59 | 0.85 | 0.75 |
| Proposed Model | 1.73 | 0.42 | 0.80 | 0.71 |

**Table 5** Comparison of the quantitative scores of several models on the MIT1003 (*Judd et al., 2009*) dataset.

| Model | NSS | SIM | AUC-Judd | AUC-Borji |
|---|---|---|---|---|
| ITTI | 1.10 | 0.32 | 0.77 | 0.76 |
| AIM | 0.82 | 0.27 | 0.79 | 0.76 |
| Judd Model | 1.18 | 0.42 | 0.81 | 0.80 |
| GBVS | 1.38 | 0.36 | 0.83 | 0.81 |
| Mr-CNN | 1.36 | 0.35 | 0.80 | 0.77 |
| CAS | 1.07 | 0.32 | 0.76 | 0.74 |
| SalGAN | 1.31 | 0.64 | 0.78 | 0.75 |
| ML-NET | 1.64 | 0.35 | 0.82 | – |
| Proposed Model | 1.35 | 0.44 | 0.88 | 0.78 |

**Table 6** Comparison of the quantitative scores of several models on the DUT-OMRON (*Yang et al., 2013*) dataset.

| Model | NSS | SIM | AUC-Judd | AUC-Borji |
|---|---|---|---|---|
| ITTI | 3.09 | 0.53 | 0.83 | 0.83 |
| AIM | 1.05 | 0.32 | 0.77 | 0.75 |
| GBVS | 1.71 | 0.43 | 0.87 | 0.85 |
| CAS | 1.47 | 0.37 | 0.80 | 0.79 |
| Proposed Model | 1.84 | 0.45 | 0.88 | 0.76 |

SIM and AUC-Judd metrics, while the GBVS model provides the best results for the NSS metric. Finally, Table 6 shows that, with the DUT-OMRON dataset, the proposed model achieved the best result in terms of the AUC-Judd metric, while the GBVS model is the best in terms of the AUC-Borji metric.

**Table 7  Runtime of the proposed model and ten visual saliency models.**

| Model | Training | Deep Learning | Run Time |
|---|---|---|---|
| BMS | No | No | 0.3 S |
| CAS | No | No | 16 S |
| GBVS | No | No | 2 S |
| ITTI | No | No | 4 S |
| Mr-CNN | yes | Yes | 14 S (GPU) |
| SalNet | yes | Yes | 0.1 S (GPU) |
| eDn | yes | Yes | 8 S (GPU) |
| AIM | yes | No | 2 S |
| Judd Model | yes | No | 10 S |
| DVA | yes | Yes | 0.1 S (GPU) |
| Proposed Model | yes | Yes | 12 S |

## Qualitative comparison of the proposed model with other advanced models

The qualitative results obtained by the proposed model are compared with five state-of-the art models, ITTI, FES, CovSal, GBVS, and SDS-GM (*Li & Mou, 2019*), on the aforementioned datasets (i.e., TORONTO, MIT300, MIT1003, and DUT-OMRON). Figure 4 shows the visual saliency map results and the proposed model visual saliency prediction, i.e., generating saliency map, within the given images. Based on the evaluation of the proposed model, the proposed model produces saliency maps comparable to other state-of-the-art models.

## Ablation study

In this work, we evaluated several different aspects of the proposed model's architecture. Table 8 illustrates the results of the experiments conducted in this work. Based on the architecture of the proposed model, we suggested 13 different scenarios in order to find an optimum architecture. Several conclusions were obtained based on these experiments:

   (1) From scenarios S1 to S4, we can see the best global accuracy is achieved with 3 encoder-3 decoder stages (i.e., global accuracy was 85.05% and loss function was 0.2384).

   (2) S7 describes the proposed model using 3 convolutional modules & 3 inception modules. This architecture also produced the best global accuracy (i.e., global accuracy was 93.63%, and loss function was 0.1051) compared to S5 and S6, which contain one and two inception modules, respectively.

   (3) S13 is the last scenario we selected as the entire model, including 3 convolutional, 3 inception, and 1 residual module (i.e., Fig. 1). This scenario produced a higher global accuracy (i.e., global accuracy was 97.05%, and loss function was 0.07) compare to those of scenarios S11 and S12.

## CONCLUSIONS

A new deep CNN model has been proposed in this paper for predicting visual saliency in the field of view. The main novelty of this model is its use of a new deep learning

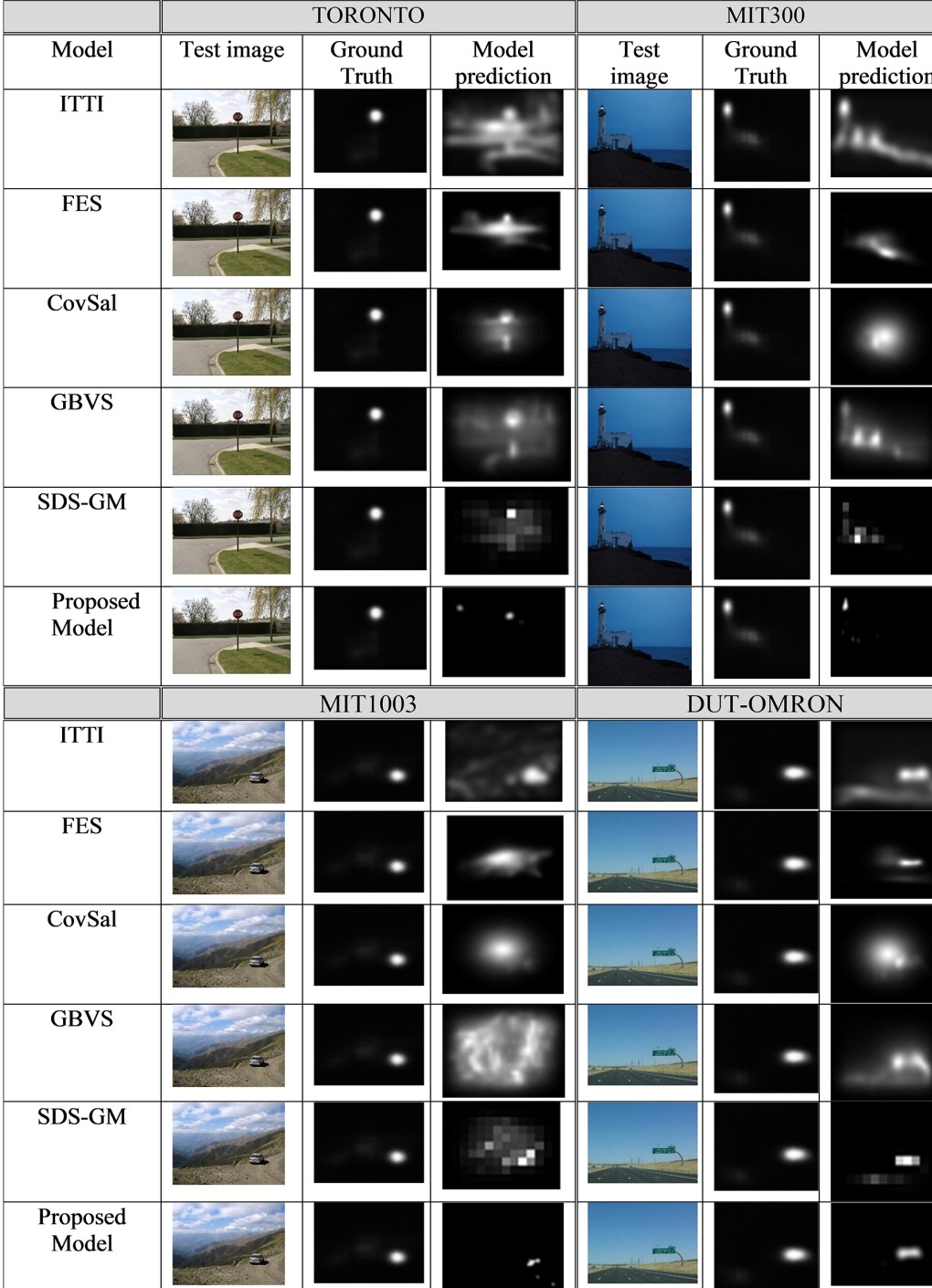

**Figure 4** The saliency maps obtained from the proposed model and five other state-of-the-art models for a sample image from the TORONTO, MIT300, MIT1003, and DUT-OMRON datasets.

**Table 8** Different FCN models applied in this study.

| Scenarios | Description | Training Accuracy | Training Loss | Validation Accuracy | Validation Loss |
|---|---|---|---|---|---|
| | **FCN Models** | **Training** | | **Validation** | |
| S1 | 2 convolutional modules | 79.14% | 0.2650 | 78.88% | 0.2700 |
| S2 | 3 convolutional modules | 85.05% | 0.2384 | 83.08% | 0.2571 |
| S3 | 4 convolutional modules | 83.47% | 0.2548 | 82.94% | 0.2608 |
| S4 | 5 convolutional modules & | 80.04% | 0.2873 | 76.52% | 0.2775 |
| S5 | 3 convolutional modules & 1 inception modules | 89.69% | 0.2119 | 85.05% | 0.2231 |
| S6 | 3 convolutional modules & 2 inception modules | 90.84% | 0.1995 | 85.37% | 0.2454 |
| S7 | 3 convolutional modules & 3 inception modules | 93.63% | 0.1051 | 89.24% | 0.1666 |
| S8 | 3 convolutional modules & 1 residual modules | 87.55% | 0.2138 | 84.97% | 0.2317 |
| S9 | 3 convolutional modules & 2 residual modules | 83.23% | 0.2597 | 82.10% | 0.2684 |
| S10 | 3 convolutional modules & 3 residual modules | 81.66% | *0.2750* | 79.12% | 0.2921 |
| S11 | 3 convolutional modules & 1 inception module & 1 residual module | 89.46% | 0.1829 | 88.59% | 0.1889 |
| S12 | 3 convolutional modules & 2 inception module & 1 residual module | 92.73% | 0.1255 | 89.92% | 0.2111 |
| S13 | 3 convolutional modules & 3 inception module & 1 residual module | 97.05% | 0.07 | 90.64% | 0.1588 |

network with three encoders and three decoders (convolution and deconvolution) for visual saliency prediction, as well as its inclusion of two modules (inception and residual modules). The proposed model was trained from scratch and used the data augmentation technique to produce variations of images. The experiment results illustrate that the proposed model achieves superior performance relative to other state-of-the-art models. Moreover, we discovered that an increase in the number of training images will increase the model prediction accuracy (i.e., improvement in model performance); however, the implementation of the model requires a large amount of memory and so it is difficult to use large numbers of training images. Furthermore, because the model was trained from scratch, we expected the model will require more training data that other models, which are currently unavailable.

A promising direction for future research is to collect a new dataset, generate its ground truth, and design new models with good performance and improved evaluation metrics based on the one proposed herein. Extending the proposed model and applying it to examples of dynamic saliency (i.e., video images), is another plausible and interesting avenue of research. The proposed model may also facilitate other tasks, such as scene classification, salient object detection, and object detection, making it applicable in a number of disciplines. Importantly, future models based on that proposed herein should be able to learn from high-level understanding, so they are able to, for example, detect the most important object of the image (e.g., focusing on the most important person in the room). Saliency models also need to understand high-level semantics in the visual scene (i.e., semantic gap), and cognitive attention studies can help to overcome some of the restrictions identified in the proposed model.

### Funding

Bashir Muftah Ghariba received financial support from the Libyan Ministry of Higher Education and Scientific Research, and Elmergib University, Alkhums, for the PhD program. Memorial University of Newfoundland supported the publication fee. The funders had no role in study design, data collection and analysis, decision to publish, or preparation of the manuscript.

### Grant Disclosures

The following grant information was disclosed by the authors:
Libyan Ministry of Higher Education and Scientific Research.
Elmergib University, Alkhums.
Memorial University of Newfoundland.

### Competing Interests

The authors declare there are no competing interests.

### Author Contributions

- Bashir Muftah Ghariba conceived and designed the experiments, performed the experiments, analyzed the data, performed the computation work, prepared figures and/or tables, authored or reviewed drafts of the paper, and approved the final draft.
- Mohamed S Shehata conceived and designed the experiments, prepared figures and/or tables, authored or reviewed drafts of the paper, and approved the final draft.
- Peter McGuire conceived and designed the experiments, prepared figures and/or tables, and approved the final draft.

### Data Availability

The raw data and code are available at:

- Bylinskii et al., (2012): ''MIT Saliency Benchmark''. MIT. Dataset. http://saliency.mit.edu/results_mit300.html.

- *Ghariba, Shehata & McGuire (2019)*: ''Saliency-_model_-2019''. Github. Code. https://github.com/Bashir2020/Saliency-_model_-2019.

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
