# Peer review of "A novel fully convolutional network for visual saliency prediction"

_PeerJ Computer Science, doi:10.7717/peerj-cs.280_

## Round 0.1 · original submission · Major Revisions

Dear authors,

Reviewers suggest that (1) the contribution should be stated more clearly, with better explanations and figures, and (2) the comparisons with SoA studies on benchmark datasets are needed.

Please consider the detailed reviewer comments to prepare a revision for your paper.

Reviewer 1 ·

Basic reporting

- The paper is well written. The language is clean and understandable.
- The referenced works are insufficient. For example: "Deep Gaze I: Boosting Saliency Prediction with Feature Maps Trained on ImageNet", "A Deep Multi-Level Networkfor Saliency Prediction", "Predicting Eye Fixations using Convolutional Neural Networks ", "Large-Scale Optimization of Hierarchical Features for Saliency Prediction in Natural Images" papers are related this work too. Some of them also contains multi-scale inference as in the proposed work.
- The tables are readable but the given figures are not. Figure 1, 2 and 3 are not clear and zooming makes them worse. It would be better to fix them.
- Both Table 4 and 5 belong to MIT1003 dataset. MIT300 experiment is missing.

Experimental design

- The authors propose a deep bottom-up saliency model but using "This testing revealed that the most salient objects in the image are detectable by the proposed model." , "...proposed model clearly predicts most of the salient objects within the given images." statements misguide the readers by making them think if it will be compared with salient object detection methods or analyzing the relation between the bottom-up saliency and objectness. The proposed model is a bottom-up saliency model not salient object detection. The authors should fix the mentioned statements to avoid confusion or they should propose a salient object method and compare their results with the related salient object detection methods.
- The methods in comparison tables are not sufficient. It does not contain the state-of-the-art methods as SalGAN, Deep Gaze I - II, ML-NET etc. To understand the contribution of the proposed method, the comparison tables should include the mentioned models' results.
- MIT300 results are missing.
- The results are not analyzed properly. The reasoning behind the given scores is not explained.

Validity of the findings

- The results are not discussed.
- The motivation is not clear. The experiments in Table 7 is well-thought but it gives the impression that the authors decide to model based on the experimental results only instead of a scientific reason.

Reviewer 2 ·

Basic reporting

'no comment'

Experimental design

'no comment'

Validity of the findings

'no comment'

Additional comments

This paper proposes a new deep learning model based on a Fully Convolutional Network (FCN) architecture. The proposed model is trained in an end-to-end style and designed to predict visual saliency. The model is based on the encoder-decoder structure and includes two types of modules. The proposed video saliency detaction idea is interesting. However, the descriptions of introduction and methods could be further improved, and some important experiments should be added. Many major confused concerns should be modified and clarified as follows.
1) The abstract is too long and redundant. The authors had better write the part of the method clearly and concisely.
2) In Section 1. Introduction, the authors lack a sufficiently comprehensive review of the related work. The authors should add a new subsection “Related works”, and they should they should carefully introduce these related works about saliency detection as follows:
*Saliency-aware video object segmentation, IEEE TPAMI
* Inferring salient objects from human fixations, IEEE TPAMI
*Correspondence driven saliency transfer, IEEE TIP
* Deep visual attention prediction, IEEE TIP
* Video Saliency Detection Using Object Proposals, IEEE TCYB
* Revisiting video saliency prediction in the deep learning era, IEEE TPAMI
* Video Salient Object Detection via Fully Convolutional Networks, IEEE TIP
3) In Section 2. Material and methods, Figure 1. Architecture of the proposed model should be reproduced, and the authors should add more details to explain the architectures of their network.
4) In Section 3. Experimental results, the authors should add more training and implementation details for better presentation.
5) In Section 3. Experimental results, the comparison results are not enough. The authors should add more experimental comparisons on more benchmarks (like [ref1] Revisiting video saliency prediction in the deep learning era, IEEE TPAMI).
6) The computational speed of the proposed algorithm is not mentioned, it is better to add one table to list the computational statistics with the state-of-the-art methods.
7) There are still some typos and grammar errors, and the authors should do a more careful proof-reading for the reversion.

---

## Round 0.2 · Minor Revisions

Please consider the comments of the reviewer for the additional experiments required and go over your manuscript for editing.

Reviewer 1 ·

Basic reporting

The improvements are very good. There are some typos in the manuscript. Some of them: In introduction section: "... human eye movement and visual attention [8]" reference should be fixed, "...performance of Visual saliency" should be "visual saliency", "Semantic Segemnation ", "34. Experimental results " section "...are described in detial." should be fixed. The authors should review the manuscript for other typos.

Experimental design

According to the MIT300 results, SALGAN, ML-NET outperforms the proposed approach. The authors should include these methods for other datasets, too. SalGAN and ML-NET implementations are currently avaliable, the authors should generate results for other datasets by using them. Also ML-NET provides generated saliency maps for MIT1003 dataset, which authors should definitely include this results.
The analysis of each module is beneficial to observe their contribution to performance. But the authors should do a fair comparison as mentioned previously.

Validity of the findings

No comment

---

## Round 0.3 · accepted · Accept

Dear authors,

Thanks for your review. The final revision is accepted for publication.
Congratulations